# Using a quantum work meter to test non-equilibrium fluctuation theorems

Federico Cerisola [1,2], Yair Margalit[3], Shimon Machluf[4], Augusto J. Roncaglia[1,2], Juan Pablo Paz[1,2] & Ron Folman[3]

Work is an essential concept in classical thermodynamics, and in the quantum regime, where the notion of a trajectory is not available, its definition is not trivial. For driven (but otherwise isolated) quantum systems, work can be defined as a random variable, associated with the change in the internal energy. The probability for the different values of work captures essential information describing the behaviour of the system, both in and out of thermal equilibrium. In fact, the work probability distribution is at the core of "fluctuation theorems" in quantum thermodynamics. Here we present the design and implementation of a quantum work meter operating on an ensemble of cold atoms, which are controlled by an atom chip. Our device not only directly measures work but also directly samples its probability distribution. We demonstrate the operation of this new tool and use it to verify the validity of the quantum Jarzynksi identity.

[1] Departamento de Física, Facultad de Ciencias Exactas y Naturales, Universidad de Buenos Aires Ciudad Universitaria, 1428 Buenos Aires, Argentina. [2] Instituto de Física de Buenos Aires, CONICET-UBA, Ciudad Universitaria, 1428 Buenos Aires, Argentina. [3] Department of Physics, Ben-Gurion University of the Negev, Be'er Sheva 84105, Israel. [4] Van der Waals-Zeeman Institute, University of Amsterdam, Science Park 904, PO Box 94485, 1090 GL Amsterdam, The Netherlands. Correspondence and requests for materials should be addressed to F.C. (email: cerisola@df.uba.ar) or to J.P.P. (email: paz@df.uba.ar)

Classical fluctuation theorems establish surprising relations between non-equilibrium and equilibrium concepts. In particular, the work performed on a system during non-equilibrium processes is connected with key concepts of equilibrium thermodynamics, such as the free-energy[1, 2]. These relations have been verified in various experiments involving microscopic thermodynamic systems[3–5]. Recent advances in quantum technologies enable the control of small quantum systems that can be manipulated far from the regime where the usual thermodynamical laws are obeyed. This triggered the development of the rapidly growing field of non-equilibrium quantum thermodynamics[6–9].

When quantum fluctuations dominate, defining and measuring work and heat, two central concepts in classical thermodynamics, is non-trivial. For driven, but otherwise isolated, quantum systems, work $w$ is a random variable associated with the change in the internal energy[10], as the first law of thermodynamics indicates. Thus, the commonly accepted definition of quantum work requires a two-time measurement strategy, which consists of performing two projective energy measurements, one at the beginning and the other at the end of the process. Then, work is associated with the measured energy difference. However, implementing the two-time measurement is experimentally difficult[11, 12] due to the fact that the two projective measurements are unavoidably disruptive (see ref. [13] for an ion trap implementation). Alternative methods to evaluate the work probability

distribution that rely on the direct estimation of its Fourier transform were also proposed in refs. [14, 15] and later implemented in NMR experiments[16].

In this paper we present the design and the experimental implementation of a "quantum work meter" (QWM) operating on an ensemble of cold atoms, combining the idea presented in ref. [17] and the experimental setup used in ref. [18]. Our QWM is conceptually different from previous work-measurement devices. Its main advantage is that the QWM efficiently samples $P(w)$, which is a direct observable in the experiment. Namely, our QWM not only directly measures work but also directly samples its probability distribution $P(w)$ (i.e. the outcome $w$ is obtained with probability $P(w)$). As the work probability distribution plays a central role in the fluctuation theorems of non-equilibrium quantum thermodynamics, the QWM is an ideal tool to test their validity. In particular, we use it to verify the Jarzynski identity[1, 10, 19–21].

## Results

**Work measurement and the QWM.** A QWM is an apparatus that measures the work performed on a driven quantum system whose Hamiltonian varies from an initial $H$ to a final $\tilde{H}$ with eigenvalues $E_n$ and $\tilde{E}_m$, respectively. For an isolated system $S$, with a $D$-dimensional space of states, the number of different values of work is bounded by $D^2$. Therefore, the pointer of the QWM has

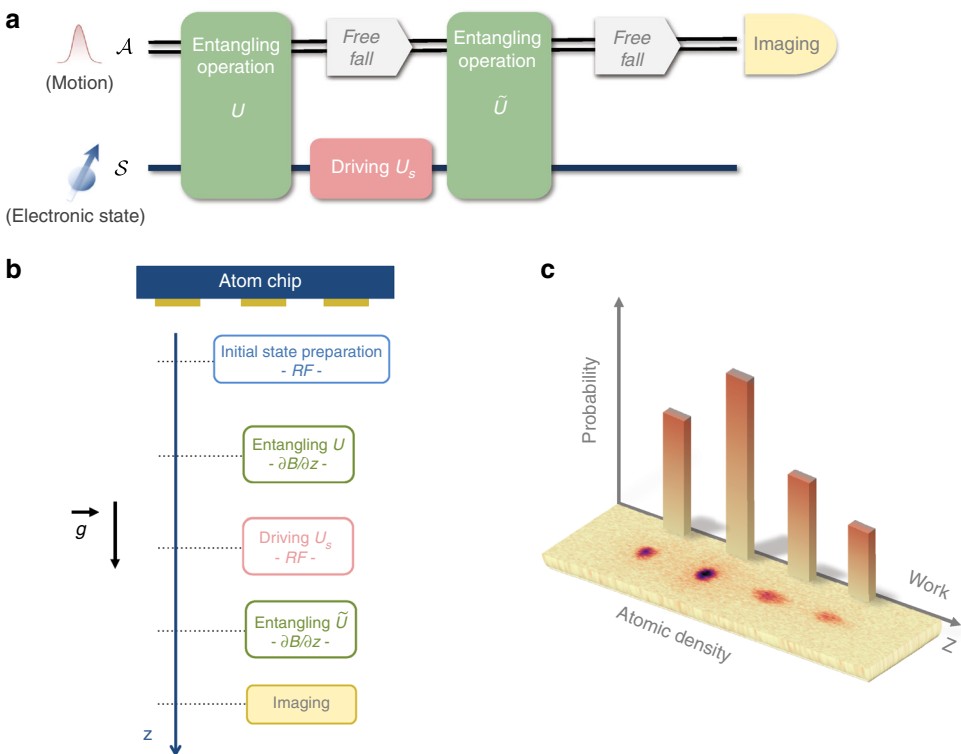

**Fig. 1** The quantum work meter. **a** A quantum circuit for the quantum work meter (QWM). $S$ and $A$ are entangled so that the eigenvalue of the observable $H$ of the system $S$ is coherently recorded by $A$. Then $S$ is driven by $U_S$. Finally, another entangling operation between $S$ and $A$ creates a record of $w$ on $A$. In the experiment, $A$ is encoded in the motional degree of freedom of the atoms along the vertical direction $z$, which also evolves while freely falling. $S$ is the pseudospin associated with the Zeeman sub-levels of a $^{87}$Rb atom. **b** Physical operations for the QWM on an atom chip: (i) The atoms, prepared in state $|2\rangle$, are released from the trap, and a RF field generates an initial pseudo-thermal state. (ii) After 2.4 ms, internal and motional degrees of freedom are entangled with a magnetic gradient pulse ($U$), applied for a duration of $\tau = 40\,\mu$s. (iii) Another RF field drives $S$. (iv) 3.1 ms after the application of $U$, a second magnetic gradient pulse ($\tilde{U}$) is applied for a duration of $\tilde{\tau} = 300\,\mu$s. At this stage, $A$ keeps a record of the different work values. (v) After 18.2 ms from the application of $\tilde{U}$, the positions and optical densities of the atomic clouds are measured. The number of atoms in each cloud reveals the work probability in a single experimental realisation. **c** Image of the four clouds obtained at the end of a single run of the QWM. The four possible values of $w$ fix the position of each cloud

$D^2$ distinct positions (one for each value of $w = w_{nm} = \tilde{E}_m - E_n$). The QWM presented here enables us to choose $H$ and $\tilde{H}$ (fixing the possible values of $w$) and to vary the intermediate driving (inducing different evolution operators denoted as $\mathcal{U}_\mathcal{S}$). In this way, we vary the probability $P(w)$, which depends on the intermediate driving $\mathcal{U}_\mathcal{S}$.

By sampling $P(w)$, we use the QWM to verify a fundamental result in non-equilibrium quantum thermodynamics: the Jarzynski identity. This identity states that for any initial state with populations identical to the ones associated to a thermal Gibbs state and for any distribution $P(w)$, the linear combination $\langle e^{-\beta w} \rangle = \sum_w e^{-\beta w} P(w)$, where $\beta = 1/k_B T$ is the inverse temperature of the system, is an equilibrium property (rather than a non-equilibrium one). The Jarzynski identity (see Supplementary Note 1) reads

$$\langle e^{-\beta w} \rangle = e^{-\beta \Delta F}, \tag{1}$$

where $\Delta F$ is the free energy difference between the thermal states associated with the Hamiltonians $H$ and $\tilde{H}$. In the absence of degeneracies, this implies that the vector formed by the $D^2 - 1$ measured probabilities belongs to a $D^2 - 2$ dimensional hyperplane: the 'Jarzynski manifold' (as shown in Supplementary Note 1, further constraints restrict this dimensionality to $(D-1)^2$). With the QWM we measure $P(w)$ for different driving fields showing that all probability vectors belong to the same manifold. By characterising this manifold, we not only verify the identity but also independently estimate the free energy difference $\Delta F$[1,3–5].

The work distribution sampled by the QWM[10, 19–21] is:

$$P(w) = \sum_{n,m} p_n p_{m|n} \, \delta[w - (\tilde{E}_m - E_n)]. \tag{2}$$

Thus, $P(w)$ is the probability density of finding the energy difference $w$ after a measurement of $H$ followed by an intermediate driving $\mathcal{U}_\mathcal{S}$ and a final measurement of $\tilde{H}$. This is indeed the case if $p_n$ is the probability of obtaining $E_n$ when measuring $H$ and $p_{m|n}$ is the probability of obtaining $\tilde{E}_m$ when measuring $\tilde{H}$ given that $E_n$ was detected at the beginning. Equation (2) defines a probability density that is independent of the initial coherences in the energy basis. For the discrete $D^2$ values of $w$ we will use $P(w)$ to denote the probability (not the density) of each $w$. The concept on which our QWM is based was first discussed in refs. [17, 22] where it was noticed that the work done on $S$, can be detected by performing a generalised quantum measurement, which enables the number of outcomes to be larger than $D$. This can be done by entangling $\mathcal{S}$ with an ancilla $\mathcal{A}$ that stores a coherent record of $w$. Then a standard measurement on $S$ can reveal $w$. Similar strategies have been later studied and extended to other contexts in refs. [22–24].

**Design and operation of the QWM.** A pictorial representation of the protocol we follow to operate the QWM is shown in Fig. 1a. The QWM is designed to measure the work done on a system $\mathcal{S}$ whose Hamiltonian changes from $H$ to $\tilde{H}$ and which is subjected to a driving $\mathcal{U}_\mathcal{S}$ in between. We couple $\mathcal{S}$ to a continuous variable system $\mathcal{A}$ and use $\hat{z}_\mathcal{A}$ to denote its position (the generator of translations along the momentum $p$). A coherent record of $w$ is created by an 'entangling interaction' between $\mathcal{A}$ and $\mathcal{S}$ that must take place before and after the driving $\mathcal{U}_\mathcal{S}$. The unitary operators representing these interactions are: $U = e^{-i\lambda \hat{z}_\mathcal{A} \otimes H/\hbar}$ and $\tilde{U} = e^{i\lambda \hat{z}_\mathcal{A} \otimes \tilde{H}/\hbar}$, where $\lambda$ is a coupling parameter. Thus, $U$ and $\tilde{U}$ respectively translate $\mathcal{A}$ along $p$ by a displacement proportional to $(-\lambda H)$ and $\lambda \tilde{H}$. Then, as shown in detail in Supplementary Note 3, the final measurement of $p$ on $\mathcal{A}$ yields a random result whose

distribution $P_\mathcal{A}(p)$ is a smeared version of the true work distribution $P(w)$ defined in Eq. (2). In fact, outcome $p$ is obtained with a probability density $P_\mathcal{A}(p) = \int dw P(w) f(p - \lambda w)$, where the window function $f(p) = |\langle p|\phi \rangle|^2$ is fixed by $|\phi\rangle$, the initial state of $\mathcal{A}$ (thus, by localising $|\phi\rangle$ we improve the accuracy in the estimation of $P(w)$).

A 'universal' QWM is an apparatus which can measure $w$ and sample $P(w)$ for any possible choice of $H$ and $\tilde{H}$. To build it, we need enough control to enforce the entangling operators $U$ and $\tilde{U}$ for any choice of $H$ and $\tilde{H}$. Remarkably, this is achieved for a 2-level system by the atom chip implementation we describe below.

**Experimental implementation of the QWM.** To describe our QWM we should identify the physical systems representing $\mathcal{S}$ and $\mathcal{A}$, the way in which $H$ and $\tilde{H}$ can be chosen, and how the associated $U$ and $\tilde{U}$ are implemented. In our experiment we represent $\mathcal{S}$ by the subspace associated with the Zeeman sublevels $|1\rangle \equiv |F = 2, m_F = 1\rangle$ and $|2\rangle \equiv |F = 2, m_F = 2\rangle$ of a $^{87}$Rb atom that, as in ref. [18] behaves as a two-level system (see below). The motional degree of freedom of the atom plays the role of $\mathcal{A}$.

A key element of the QWM presented here is the atom chip[25], which efficiently entangles the internal and motional degrees of freedom of an atom just ~100 µm away from the atom chip surface, through short and strong Stern-Gerlach type magnetic gradient pulses. These pulses are generated using a 3-current-carrying wire setup on the atom chip surface (described in ref. [26] and the Methods section). A gradient pulse along the $z$ direction with amplitude $B'$ and duration $\tau$, induces a momentum kick $m_F \delta p$ on an atom in the $m_F$ state ($\delta p \sim \mu_B g_F B' \tau$, where $\mu_B$ and $g_F$ are, respectively, the Bohr magneton and the Landé factor[18]). The evolution of the state of the atom induced by such a pulse is described by the unitary operator $U_p = e^{i\delta p \hat{z}_\mathcal{A} \otimes \hat{\sigma}/\hbar}$, where $\hat{\sigma} = |1\rangle\langle 1| + 2|2\rangle\langle 2|$. This physical operation translates $\mathcal{A}$ along the momentum $p$ by a displacement $\delta p \hat{\sigma}$ (notice that the operator $\hat{\sigma}$ defines the magnetic dipole moment of the atom since $\hat{\sigma} = \sum_{m=1,2} m|m\rangle\langle m|$). As described below, we apply two gradient pulses with different amplitudes ($B'$ and $\tilde{B}'$) and different durations ($\tau$ and $\tilde{\tau}$). Thus, defining $H = E\hat{\sigma}$ and $\tilde{H} = \tilde{E}\hat{\sigma}$, $U_p$ and $\tilde{U}_p$ implement the required entangling operation $U$ and $\tilde{U}$, respectively. In this implementation $\lambda$ is consequently replaced by $-\delta p/E$ and $\delta \tilde{p}/\tilde{E}$, enforcing $\tilde{E}/E = -\delta \tilde{p}/\delta p$. The momentum kicks induced by both pulses are controlled in the experiment, and consequently, by fixing their ratio, we can simulate an arbitrary system with initial and final Hamiltonians $H$ and $\tilde{H}$ which are characterised by $\tilde{E}/E$ having the same ratio. Finally, let us note that the two pulses utilise $B'$ and $\tilde{B}'$ with opposite signs to ensure that the sequence creates a record of work corresponding to $\tilde{E}_m - E_n$.

To achieve universality we only need to be able to fix the energy splitting $E$ and $\tilde{E}$ of $H$ and $\tilde{H}$, as well as their eigenbasis. The traces of $H$ and $\tilde{H}$ (the sum of their eigenvalues) do not affect $P(w)$ but only add a constant to all values of $w$. As arbitrary $E$ and $\tilde{E}$ can be simulated and any change of basis can be absorbed into $\mathcal{U}_\mathcal{S}$, we conclude that our atom chip QWM can sample $P(w)$ for an arbitrary 2-level system and is thus universal.

The $^{87}$Rb atoms are magnetically trapped in state $|2\rangle$ and evaporatively cooled to a Bose–Einstein condensation (BEC). The BEC is released from the trap and a radio-frequency (RF) pulse is used to prepare a superposition of $|1\rangle$ and $|2\rangle$. A strong homogeneous magnetic field (created by external coils) suppresses the transitions taking $|1\rangle$ into the $|2,0\rangle$ state (due to the non-linear Zeeman effect[18]). The initial populations ($p_1$ and $p_2$) are chosen so that the temperature is determined via $\beta E = \ln$

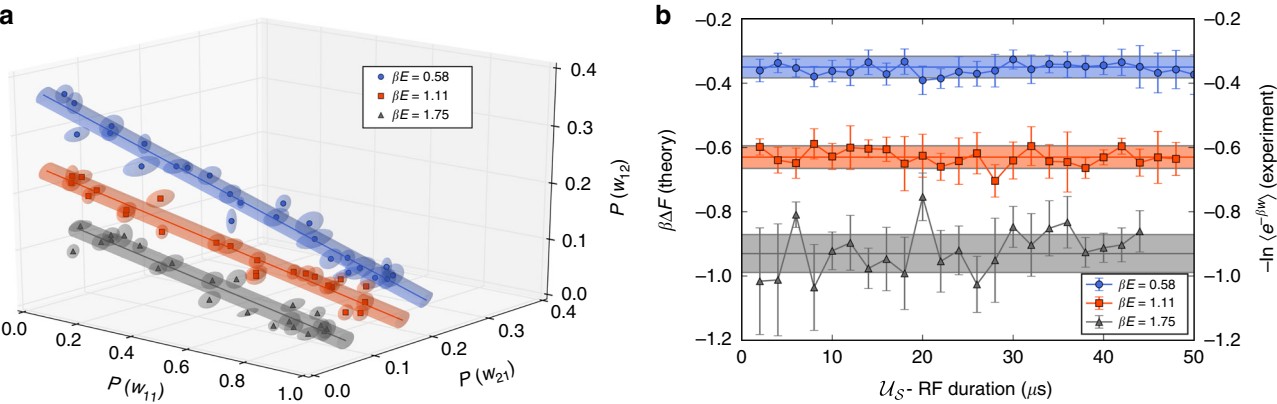

**Fig. 2** The Jarzynski identity. **a** Each point defines a probability vector (with its experimental error) measured for a certain driving. The three lines correspond to three temperatures: $\beta E = 0.58 \pm 0.02$ (blue circle), $1.11 \pm 0.02$ (red square) and $1.75 \pm 0.04$ (grey triangle). For each temperature all points lie in the same Jarzynski manifold (which in this case is a line). Reported errors are the SEM of three independent experiments with the same initial parameters and driving. The projections onto the three different axes of the probabilities are shown in detail in Supplementary Fig. 2. **b** $-ln\langle e^{-\beta w}\rangle = -ln\left[\sum_w e^{-\beta w}P(w)\right]$ becomes independent of the duration of the intermediate driving (for three temperatures). The dots are the calculated values using the measured work distribution in the Jarzynski identity, and the solid line is the theoretical estimate of $\beta\Delta F$ (with an uncertainty due to the uncertainties in the temperature and energy splitting). Error bars are the SEM

**Table 1 Estimates of $\beta\Delta F$ and $\Delta F$ for three different temperatures**

| $\beta E$ | $\beta\Delta F$ (JI) | $\beta\Delta F$ (PF) | $\Delta F/E$ (JI) | $\Delta F/E$ (PF) |
|---|---|---|---|---|
| $0.58 \pm 0.02$ | $-0.36 \pm 0.04$ | $-0.35 \pm 0.03$ | $-0.62 \pm 0.07$ | $-0.60 \pm 0.06$ |
| $1.11 \pm 0.02$ | $-0.63 \pm 0.05$ | $-0.63 \pm 0.04$ | $-0.57 \pm 0.05$ | $-0.57 \pm 0.04$ |
| $1.75 \pm 0.04$ | $-0.92 \pm 0.09$ | $-0.93 \pm 0.06$ | $-0.53 \pm 0.05$ | $-0.53 \pm 0.04$ |

We show the estimation obtained using the Jarzynski identity (JI) and from a direct calculation of the partition function (PF)

$(p_1/p_2)$. The initial motional state is a wave-packet localised in position and momentum.

It should be noted that the initial internal state of the atom, while having the same populations as defined by the temperature of a thermal state, is still a pure state. However, the quantum coherences of this initial state do not affect the results of the QWM. As explained in Supplementary Note 3, the contribution of the initial coherences to the final probability is multiplied by the overlap between the motional states of the atom associated with the different values of work. Thus, when the atomic clouds associated with the different work values are well separated, the effect of initial coherences is negligible. In this regime, our experiment gives the same result as the one we would obtain by preparing an initial thermal state (with no coherences). The study of the importance of the initial coherences in the definition of work is an interesting topic in itself, which is beyond the scope of our paper (see, for example, refs. [24, 27–30]).

The experimental sequence, presented in Fig. 1b, is: (i) prepare the initial state and release the cloud (which then freely falls along $z$, the direction of gravity), (ii) apply the magnetic gradient $U$ along $z$, (iii) apply the driving $\mathcal{U}_S$ by exposing the atoms to a RF field resonant with the Zeeman splitting induced by the homogeneous bias field, (iv) apply the gradient $\tilde{U}$, (v) obtain an image of the four clouds after a time-of-flight and count the number of atoms in each cloud. More details of the experiment can be found in the Methods section and Supplementary Note 3. For the experimental demonstration presented here we set the ratio between the measured momentum kicks induced by the two pulses to $-\delta\tilde{p}/\delta p = 0.56 \pm 0.02$. Hence, our realisation of the QWM samples the work distribution of a simulated system in which the energy splitting is reduced to 56% of its original value, from $E$ to $\tilde{E}$, while driven by $\mathcal{U}_S$.

Figure 1c shows a typical image obtained by the QWM. Four clouds are visible. From the positions of the centre of each cloud, $\bar{z}$, we infer the total momentum shift, $\bar{p}$, induced by the pulses on that cloud (we take into account both the free fall and the kicks induced by the pulses, see Supplementary Note 3). Then, we obtain the corresponding value of work as $w = E\bar{p}/\delta p$ ($w$ is proportional to $E$, whose value, together with the measurement of the remaining quantities, determine the work $w$). Furthermore, the probability $P(w)$ for each $w$ is directly measured by the number of atoms in each cloud. Notably, this experiment determines the entire $P(w)$ distribution in a single shot.

**Testing the Jarzynski identity.** We repeat the experiment fixing the timing, duration and pulse strength. We consider three initial $\beta$'s and vary the intermediate driving $\mathcal{U}_S$ by changing the duration of the RF field. In this way, we obtain many sets of probability distributions, each of which defines a 3D-vector (as there are three independent probabilities). When we represent all these vectors in the same 3D-plot, we find that they all belong to the same $\beta$-dependent manifold. Figure 2a shows that this manifold is a $\beta$-dependent line (the dimensionality of this 'Jarzynski manifold' is $(D-1)^2$, which in this case equals 1).

Using the measured work probabilities we calculate the exponential average of the work $\langle e^{-\beta w}\rangle$ for each driving field. Figure 2b displays the value of $G = -ln\left[\langle e^{-\beta w}\rangle\right] = -ln\left[\sum_w e^{-\beta w}P(w)\right]$ as a function of the duration of the intermediate RF field, that parametrises $\mathcal{U}_S$. As established by the Jarzynski identity, $G$ is independent of the driving field and only depends on $\beta$. The horizontal lines in Fig. 2b are the theoretically predicted values of $\beta\Delta F$, obtained from a direct calculation (with its own theoretical uncertainty, due to the error

in the estimation of $\beta E$). This calculation simply involves computing the initial and final partition functions, respectively, denoted as $Z$ and $\tilde{Z}$, and using the identity $\beta\Delta F = \ln\left(Z/\tilde{Z}\right)$. We find, as the Jarzynski identity establishes, $G = \beta\Delta F$. From Fig. 2b one can notice that the largest errors in the estimation of $\beta\Delta F$ appear for $\beta E = 1.75$. In this case, $P(w) \lesssim 0.1$ for two values of $w$ and, therefore, the relative error in the atom number estimation is large, inducing a larger error in the estimation of $\beta\Delta F$.

In Table 1 we compare the measured and estimated values of $\beta\Delta F$. The uncertainty in the estimation of $\beta\Delta F$ and $\Delta F$ is close to 10%, which is enough to distinguish the three values of $\beta\Delta F$. On the other hand, in the case of $\Delta F$, there is a significant overlap in the measured values which does not allow to properly distinguish between the three different cases due to the error in the estimation of $\beta E$.

## Discussion

We presented and implemented a QWM, a new device directly sampling the work distribution on an ensemble of cold atoms. Our QWM can be used to simulate the behaviour of an arbitrary 2-level system. We implemented it with an atom chip and verified the Jarzynski identity over a wide range of non-equilibrium processes. This is the first experiment, and so far the only one, directly sampling $P(w)$ offering advantages and different perspectives over previous work-measurement schemes. Remarkably, in this cold atom experiment, the QWM extracts full statistical information about the work distribution in a single shot.

## Methods

**Initial state preparation**. After preparing the BEC, a homogeneous magnetic field of 36.7G (25 $h$MHz/$\mu_B$, where $h$ is Planck's constant) is used to push the transition to $|2,0\rangle$ out of resonance by ~180 kHz due to the non-linear Zeeman effect, which is larger than the power broadened driving RF field of $\mathcal{U}_S$. This ensures that the atoms behave as 2-level systems. The BEC is released from the trap and a RF pulse is used to prepare a superposition of $|1\rangle$ and $|2\rangle$. By varying the relative populations we consider three different pseudo-thermal states. The initial motional state is a wave packet $|\phi\rangle$, well localised at $z_0 = 91 \pm 1.2$ μm from the chip with momentum ~0.

**Entangling operations and measurement**. An inhomogeneous magnetic field is used to couple spin and motional degrees of freedom. This is generated by a current $I = 0.85$A in the 3-wire setup during a time $\tau$. The three parallel gold wires lie on the $x$ direction of the chip surface (Fig. 1b). They are 10 mm long, 40 μm wide and 2 μm thick. Their centres are at $y = -100, 0, 100$ μm and the same current runs through them in alternating directions ($-I, I, -I$, respectively), creating a 2D quadrupole field at $z = 98$ μm below the chip. After a time of flight of 2.4 ms the atoms are at $z \sim 119$ μm. At this point the first gradient pulse implements $U$: $\tau = 40$ μs with an amplitude of $B' \sim 95$G/mm, such that the momentum kick is along $+z$. Then, after 3.1 ms the atoms are at $\tilde{z} \sim 0.3$mm and the second gradient pulse implements $\tilde{U}$: $\tilde{\tau} = 300$ μs, $\tilde{B}' \sim -7.5$G/mm, such that the momentum kick is along $-z$ (since the gradient direction is inverted for $z > 200$ μm). The relative strengths of the spin-dependent forces sets the energy splitting of the Hamiltonians which in this case is on average $\bar{E}/E = -\delta p/\delta p = 0.56 \pm 0.02$ (this is the measured value, where the error takes into account fluctuations in the initial position of the cloud and in the gradient pulses). In between the entangling operation $\mathcal{U}_S$ is applied with a RF pulse. Finally, an image of the atomic clouds is obtained after a time-of-flight of 18.2 ms after the second gradient (the clouds are centred around $z \sim 3$ mm). The position and number of atoms of each cloud are determined. The momentum shifts of each cloud (that codifies the value of $w$) are obtained from the difference in positions between the clouds, that follow approximately classical trajectories (Supplementary Note 3).

**Uncertainties**. The main source of position error is the initial distance of the cloud from the atom chip, whose uncertainty is $\sim 1$%. This error is later translated to momentum uncertainty, since the field gradients are position dependent. The field gradients have a fractional uncertainty of $10^{-3}$ due to current fluctuations[18]. The central position of each cloud is estimated by fitting a Gaussian profile. Each work probability is estimated as a normalised sum of the measured optical density in a relevant region around the cloud, introducing a probability uncertainty (due to atom numbers uncertainty). Our ~5 μm optical resolution also induces an error in the determination of the position for each cloud. We perform three different runs for each combination of initial state population ratios and intermediate driving and

use the average values of position and probability. This gives us a position uncertainty of ~0.015 mm and a probability uncertainty of ~0.015 (standard error).

**Data availability**. The data that supports the findings of this study are available from the corresponding author upon request.

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

## Acknowledgements

F.C., A.J.R. and J.P.P. acknowledge financial support from ANPCyT (PICT 2013-0621 and PICT 2014-3711), CONICET and UBACyT. YM and RF gratefully acknowledge funding by the Israel Science Foundation, the EC Matter-Wave consortium [FP7-ICT-601180], and the German DFG through the DIP programme [FO703/2-1]. S.M. acknowledges financial support by the Foundation for Fundamental Research on Matter (FOM), which is part of the Netherlands Organisation for Scientific Research (NWO). We also thank the BGU nano-fabrication team for making available the high quality chip, and Zina Binstock for helping with the electronics.

## Author contributions

F.C., A.J.R. and J.P.P. envisioned and outlined the experiment. Y.M., S.M. and R.F. designed and performed the experiment. F.C., A.J.R., S.M. and Y.M. analysed the data. All authors contributed to the preparation of the manuscript coordinated by J.P.P. and A.J.R. The project was supervised by J.P.P. and R.F.

## Additional information

**Competing interests:** The authors declare no competing financial interests.

