## [Peer Review File · Nature Communications]

Reviewers' comments:

Reviewer #1 (Remarks to the Author):

This paper concerns the design and experimental implementation of a "quantum work meter" (QWM), which uses an atom chip to entangle the internal and translational degrees of an atom. The paper describes the general idea behind the QWM, then describes how it is used to measure work distributions for an effective two-state system. These tools are then brought to bear on a test of the Jarzynski identity. A particular feature of the QWM is that it simultaneously carries out the protocol on many copies of the system (many atoms), thereby generating the entire work distribution in one shot, so to speak.

I found the QWM to be ingenious and the paper to be well written. The topic is timely, as there is currently great interest in fundamental non-equilibrium thermodynamics, and in particular there is a strong focus on developing schemes for measuring the work performed on quantum systems. For these reasons I recommend that the manuscript be accepted for publication in Nature Communications.

I have a couple of small items that the authors may wish to address.

First, a recent paper on arXiv has some overlap with the current paper: <https://arxiv.org/abs/1705.10296> In particular, Protocol II in that paper is similar in spirit to the submitted paper, only there the system is coupled to the momentum of the measuring device, rather than to the position. (Compare the factors appearing on the right side of Eq. 1 in the arXiv paper, with the two unitaries U and U_{tilde} mentioned in the text at the bottom left of page 2 of the submitted manuscript.) The arXiv paper (Solinas et al) has quite a different focus from the submitted paper (Cerisola et al), and the arXiv paper is entirely theoretical. Nevertheless the authors may wish to cite it.

Also, in the final sentence before the Conclusions, it is written that the uncertainty in the free energy estimate "leaves a significant overlap for ΔF ". I'm not sure of the motivation for making this remark. It seems to suggest that ΔF ought to be independent of β , and that the experimental results are consistent with this independence, aside from experimental uncertainties. However, there is no a priori reason to expect ΔF to be independent of temperature. Perhaps I have simply misunderstood something trivial, but if so then I expect other readers will, as well. I suggest that the authors explain in a bit more detail the significance of the statement "leaves a significant overlap for ΔF ".

Reviewer #2 (Remarks to the Author):

In this manuscript the authors implement experimentally the observation of the work probability distribution and the Jarzynski identity on an ensemble of cold atoms. To do so, they use the method introduced in Ref.[1] that consists of entangling the system with an ancilla that is finally measured. The main contribution of this work is that they are able to use the motional degrees of freedom of the atoms as ancilla and an atom chip to entangle the internal degree of freedom to the motional one.

The manuscript is well structured and written. The results are experimentally and technically relevant; and I find the idea of using the motional degrees of freedom of the atoms as ancilla particularly nice.

I have only one concern that I would like the authors to clarify further. In case they do so satisfactorily, I will do recommend the paper for publication in Nature Communications.

According to the Methods section of the manuscript, the initial state of the atom is not a thermal state (dephased in the energy basis) but a coherent superposition of $|1\rangle$ and $|2\rangle$. This is problematic because of two reasons:

- 1) It contradicts the assumption of the fluctuation relations in which initial states need to be thermal.
- 2) There is an ongoing debate of how work can be defined when the system is initially coherent in the energy basis, since then it gets entangled with the battery (work source) during the driven evolution (the battery becomes in a superposition of having and not having supplied energy). See for instance M. Perarnau-Llobet, Phys. Rev. Lett. 118, 070601 (2017) for a recent discussion.

However, in the manuscript this point is just addressed as a side remark at the end of page 3 "... (initial coherences do not play any role, see SM)". In my opinion, this is an important point that deserves a deeper discussion and a conceptual explanation apart from a calculation in the third section of the SM.

Finally, a few of minor comments:

- I do consider the physical system used (cold atoms with internal and motional degrees of freedom) relevant enough to be briefly described in the abstract.
- The section on the experimental implementation of the QWM is quite dense. Its extension would improve its readability.
- In Fig. 2 (b), the errors of $\beta \Delta F$ corresponding to a lower temperature are larger. Is this uniquely due to the multiplicative factor β or there is an alternative source of errors at low temperatures?

Reviewer #3 (Remarks to the Author):

The authors present results on an experimental implementation of a so-called quantum work meter— a device which measures work including its probability distribution. As a platform for this device, they use a Rubidium BEC which is prepared and trapped in an atomic chip trap. A combination of entangling operations and free fall allows the authors to entangle the pseudo spin with the motion of the fall. Measurement of the final position and the density of the atomic clouds which experienced a momentum shift then allows for deducing the amount of work and its probability distribution. The authors then use this device to confirm the Jarzynski theorem, which relates the free energy between two Hamiltonians and the expectation value of the exponentiated work.

This manuscript is in very good condition and organized well. The authors not only address specialists in the field of cold atomic physics, but also the results are presented in a more general way for a broader audience. The procedure of the experiment is discussed clearly and the experimental findings are presented well.

In light of that, I recommend publication of this manuscript with only minor changes which are detailed below.

page 1, 3rd paragraph:

I suggest to remove the double negative in the sentence "... the linear combination [...] $P(w)$ is not a non-equilibrium but an equilibrium property ..." and rephrase it for easier understanding for the reader.

Fig 1, b)

It would help the reader to add the time lengths for each individual sub-sequence of the experiment in the picture and not only in the methods.

page 3, 2nd paragraph:

Can the authors add a sentence on the definition of the sigma operator? It is not clear to me where the factor of 2 in the $|2\rangle\langle 2|$ comes from.

page 3, 2nd paragraph:

The authors should rename the duration T to something else (e.g. t) since T is already used for the temperature in this manuscript.

page 4, 5th paragraph:

Sentence "... obtained from a simple calculation (with its own theoretical uncertainty, due to the error in the estimation of ...)". The authors should either define this "simple calculation" or at least refer to the supplement to clarify this statement for the reader.

Suggestion regarding Fig. 2 (a):

Since it is hard to see the distance of the points to the lines in the 3D plot, could the authors add projections of this graph in all three directions to the Supplemental Material for completeness?

Reviewer #1:

“This paper concerns the design and experimental implementation of a “quantum work meter” (QWM), which uses an atom chip to entangle the internal and translational degrees of an atom. The paper describes the general idea behind the QWM, then describes how it is used to measure work distributions for an effective two-state system. These tools are then brought to bear on a test of the Jarzynski identity. A particular feature of the QWM is that it simultaneously carries out the protocol on many copies of the system (many atoms), thereby generating the entire work distribution in one shot, so to speak.

I found the QWM to be ingenious and the paper to be well written. The topic is timely, as there is currently great interest in fundamental non-equilibrium thermodynamics, and in particular there is a strong focus on developing schemes for measuring the work performed on quantum systems. For these reasons I recommend that the manuscript be accepted for publication in Nature Communications.”

We thank the referee for the positive remarks.

“I have a couple of small items that the authors may wish to address. First, a recent paper on arXiv has some overlap with the current paper: <https://arxiv.org/abs/1705.10296> In particular, Protocol II in that paper is similar in spirit to the submitted paper, only there the system is coupled to the momentum of the measuring device, rather than to the position. (Compare the factors appearing on the right side of Eq. 1 in the arXiv paper, with the two unitaries U and U tilde mentioned in the text at the bottom left of page 2 of the submitted manuscript.) The arXiv paper (Solinas et al) has quite a different focus from the submitted paper (Cerisola et al), and the arXiv paper is entirely theoretical. Nevertheless the authors may wish to cite it.”

We were not aware of the paper mentioned by the Referee (that paper appeared in the arXiv after ours was submitted to Nature). In fact, the paper has some overlap with ours (the corrections in the mean value of work arising from a weak measurements of the energy difference is the focus of such paper). We briefly discussed this issue in the supplementary material of the first version of our paper. We now added a reference to this new paper (together with a reference to another related paper in Phys. Rev. E 93, 022131 (2016)). We thank the referee for pointing this out.

“Also, in the final sentence before the Conclusions, it is written that the uncertainty in the free energy estimate “leaves a significant overlap for ΔF ”. I’m not sure of the motivation for making this remark. It seems to suggest that ΔF ought to be independent of beta, and that the experimental results are consistent with this independence, aside from experimental uncertainties. However, there is no a priori reason to expect ΔF to be independent of temperature. Perhaps I have simply misunderstood something trivial, but if so then I expect other readers will, as well. I suggest that the authors explain in a bit more detail the significance of the statement “leaves a significant overlap for ΔF ” ”

We modified the text to better explain this issue. Of course, we did not intend to say that ΔF is independent of the temperature. In fact, this is not the case: the theoretical values of ΔF shown in the paper are temperature dependent. We simply wanted to say that the experimentally estimated values of ΔF , with their corresponding error bars, have some overlap for the three temperatures we considered. This is due to the error in the estimation of the initial temperature itself. We clarified this issue in the revised version.

Reviewer #2:

“In this manuscript the authors implement experimentally the observation of the work probability distribution and the Jarzynski identity on an ensemble of cold atoms. To do so, they use the method introduced in Ref.[1] that consists of entangling the system with an ancilla that is finally measured. The main contribution of this work is that they are able to use the motional degrees of freedom of the atoms as ancilla and an atom chip to entangle the internal degree of freedom to the motional one. The manuscript is well structured and written. The results are experimentally and technically relevant; and I find the idea of using the motional degrees of freedom of the atoms as ancilla particularly nice.”

We thank the referee for the positive remarks.

“I have only one concern that I would like the authors to clarify further. In case they do so satisfactorily, I will do recommend the paper for publication in Nature Communications.

According to the Methods section of the manuscript, the initial state of the atom is not a thermal state (dephased in the energy basis) but a coherent superposition of $|1\rangle$ and $|2\rangle$. This is problematic because of two reasons:

1) It contradicts the assumption of the fluctuation relations in which initial states need to be thermal.

2) *There is an ongoing debate of how work can be defined when the system is initially coherent in the energy basis, since then it gets entangled with the battery (work source) during the driven evolution (the battery becomes in a superposition of having and not having supplied energy). See for instance M. Perarnau-Llobet, Phys. Rev. Lett. 118, 070601 (2017) for a recent discussion. However, in the manuscript this point is just addressed as a side remark at the end of page 3 "... (initial coherences do not play any role, see SM)". In my opinion, this is an important point that deserves a deeper discussion and a conceptual explanation apart from a calculation in the third section of the SM.*

In the revised version (both in the main text and the SM) we clarified the role of initial coherences. Let us explain here in detail what is the issue.

The Jarzynski identity is valid if the work distribution $P(w)$ is given by Eq (2) of our manuscript, provided that the initial populations are identical to the ones associated with a thermal Gibbs state, i.e., $p_n = \exp(-\beta E_n)/Z$. In fact, the validity of the Jarzynski identity does not require the initial state to be thermal but only to have the above populations. As explained in the text, if one measures the energy at two times (the 2-time measurement strategy), the result $w_{n,m} = \tilde{E}_m - E_n$ is obtained with probability $P(w_{n,m})$. Again, the only condition for this to happen is that the initial populations are identical to the ones associated with a thermal Gibbs state. Indeed, coherences present in the initial state play no role in the probability of outcomes of the 2-time measurement strategy. This is simply because the initial energy measurement destroys them.

Our measurement strategy, presented in Ref [17] (in the revised version), produces an outcome w with a probability $P_A(w)$, which is related with the work probability $P(w)$. The relation between the sampled probability $P_A(w)$ and the work probability $P(w)$ is explained in the text and the SM but it is worth reviewing it here. For an initial thermal Gibbs state (i.e. $\rho_0 = \exp(-\beta H)/Z$), $P_A(w)$ can be rigorously shown to be a smeared version of $P(w)$ (with a smearing function determined by the initial state of the ancilla). However, if the initial state of the system is pure the relation between $P_A(w)$ and $P(w)$ may be more subtle: Even if the populations correspond to a thermal Gibbs state, as explained in the SM, coherences may play a role (modifying the sampled probability $P_A(w)$).

However, the effect of initial coherences may become irrelevant in an important case, which is precisely the one we consider in our paper. This can be understood as follows: When the total initial state is pure, the system and the ancilla become entangled after implementing the operations required by the QWM. Then, the total state is a sum of product states each of which can be associated with a different value of work, $w_{n,m}$, and involves a certain state of the ancilla $D_{nm}|\phi\rangle$ (where D_{nm} is a displacement operator, see SM). Thus, the role of the initial coherences becomes negligible when the ancillary states $D_{nm}|\phi\rangle$ become approximately orthogonal to each other. In our experiment this condition is met when the motional states associated with the different value of work do not overlap (i.e., when the four atomic clouds are well separated). In this case, the ancilla performs a strong measurement of the energy difference and the probability of the different values is precisely given by a smeared version of $P(w)$.

The above explanation justifies our statement that "coherences do not matter" in our implementation of the QWM. In fact, the goal of our apparatus is to sample the work probability distribution defined in Eq (2) and this goal is certainly achieved.

We acknowledge that there is an ongoing debate about the definition of work when initial coherences exist. However, this interesting debate is beyond the scope of our paper (whose goal, as stated above, is to design and implement an apparatus that efficiently samples the work probability distribution defined in Eq. (2)). For completeness, in the new version we added a short paragraph about the role of initial coherences and included a reference to the paper mentioned by the Referee (M. Perarnau-Llobet, Phys. Rev. Lett. 118, 070601 (2017) together with other related ones.

" Finally, a few of minor comments:

- I do consider the physical system used (cold atoms with internal and motional degrees of freedom) relevant enough to be briefly described in the abstract."

We mention this in the abstract and the introduction of the revised version.

" - The section on the experimental implementation of the QWM is quite dense. Its extension would improve its readability."

We modified this section in the revised version extending it to improve its readability.

" - In Fig. 2 (b), the errors of $\beta\Delta F$ corresponding to a lower temperature are larger. Is this uniquely due to the multiplicative factor β or there is an alternative source of errors at low temperatures? "

In general the errors at low temperatures are larger than at high temperatures. This is because at low temperatures the

populations of some of the clouds are really small (as it is explained in the main text and can be seen in Fig. 2(b) or the plots that we add in the revised version of the SM). In that case, the estimation of these low probabilities induce larger uncertainties than for high temperatures.

Reviewer #3:

“The authors present results on an experimental implementation of a so-called quantum work meter a device which measures work including its probability distribution. As a platform for this device, they use a Rubidium BEC which is prepared and trapped in an atomic chip trap. A combination of entangling operations and free fall allows the authors to entangle the pseudo spin with the motion of the fall. Measurement of the final position and the density of the atomic clouds which experienced a momentum shift then allows for deducing the amount of work and its probability distribution. The authors then use this device to confirm the Jarzynski theorem, which relates the free energy between two Hamiltonians and the expectation value of the exponentiated work.

This manuscript is in very good condition and organised well. The authors not only address specialists in the field of cold atomic physics, but also the results are presented in a more general way for a broader audience. The procedure of the experiment is discussed clearly and the experimental findings are presented well.

In light of that, I recommend publication of this manuscript with only minor changes which are detailed below.”

We thank the referee for the positive comments.

“page 1, 3rd paragraph: I suggest to remove the double negative in the sentence “... the linear combination [...] $P(w)$ is not a non-equilibrium but an equilibrium property ...” and rephrase it for easier understanding for the reader. ”

In the revised version we changed the wording of this sentence.

“Fig 1, b) It would help the reader to add the time lengths for each individual sub-sequence of the experiment in the picture and not only in the methods.”

We include this information in the caption of Fig. 1.

“page 3, 2nd paragraph: Can the authors add a sentence on the definition of the sigma operator? It is not clear to me where the factor of 2 in the $|2\rangle\langle 2|$ comes from.”

The operator $\hat{\sigma}$ defines the magnetic dipole of the atom. Thus, $\hat{\sigma} = \sum_{m_F=1,2} m_F |m_F\rangle\langle m_F|$ (we include this equation in the new version). As explained in the text, a magnetic gradient pulse induces a momentum kick $m_F \delta p$ on an atom in the m_F state. Since we consider atoms with $m_F = 1, 2$, the momentum kick for the $m_F = 2$ state is twice the one induced for the state with $m_F = 1$. This is the origin of the factor of 2 mentioned by the Referee, which appears in the definition of σ .

“page 3, 2nd paragraph: The authors should rename the duration T to something else (e.g. t) since T is already used for the temperature in this manuscript.”

We thank the referee for pointing this out, we rename these times as τ and $\tilde{\tau}$.

“page 4, 5th paragraph: Sentence “.. obtained from a simple calculation (with its own theoretical uncertainty, due to the error in the estimation of ..”. The authors should either define this “simple calculation” or at least refer to the supplement to clarify this statement for the reader.”

We clarified this issue in the revised version.

“Suggestion regarding Fig. 2 (a): Since it is hard to see the distance of the points to the lines in the 3D plot, could the authors add projections of this graph in all three directions to the Supplemental Material for completeness?”

We include these plots in the revised SM.

REVIEWERS' COMMENTS:

Reviewer #2 (Remarks to the Author):

I am satisfied with how the points I raised have been addressed. I do recommend the publication of the manuscript in Nature Communications.